# In Vitro Eradication of Planktonic, Saliva and Biofilm Bacteria Using Lingonberry Extract as a Photosensitizer for Visible Light Plus Water-Filtered Infrared-A Irradiation

**DOI:** 10.3390/nu15234988

**Published:** 2023-12-01

**Authors:** Mia Klein, Ali Al-Ahmad, Marie Follo, Elmar Hellwig, Kirstin Vach, Sigrun Chrubasik-Hausmann

**Affiliations:** 1Department of Operative Dentistry and Periodontology, Medical Center of the University of Freiburg, Faculty of Medicine, University of Freiburg, Hugstetter Straße 55, 79106 Freiburg, Germany; mia.klein@uniklinik-freiburg.de (M.K.); elmar.hellwig@uniklinik-freiburg.de (E.H.); 2Lighthouse Core Facility, Department of Medicine I, Medical Center of the University of Freiburg, Faculty of Medicine, University of Freiburg, Breisacher Straße 115, 79106 Freiburg, Germany; marie.follo@uniklinik-freiburg.de; 3Institute of Medical Biometry and Statistics, Faculty of Medicine and Medical Center, University of Freiburg, Stefan-Meier-Straße 26, 79104 Freiburg, Germany; kirstin.vach@uniklinik-freiburg.de; 4Institute of Forensic Medicine, Faculty of Medicine, University of Freiburg, 79104 Freiburg, Germany; sigrun.chrubasik@klinikum.uni-freiburg.de

**Keywords:** lingonberry, *Vaccinium vitis-idea*, oral pathogenic bacteria, photosensitizer, antimicrobial photodynamic treatment, visible light plus water-filtered infrared-A irradiation

## Abstract

Antimicrobial photodynamic treatment (aPDT) with visible light plus water-filtered infrared-A irradiation (VIS-wIRA) and natural single- or multi-component photosensitizers (PSs) was shown to have potent antimicrobial activity. The aim of this study was to obtain information on the antimicrobial effects of aPDT-VIS-wIRA with lingonberry extract (LE) against bacteria that play a role in oral health. Planktonic bacterial cultures of the Gram-positive *E. faecalis* T9, *S. mutans* DSM20523, *S. oralis* ATCC 35037 and *S. sobrinus* PSM 203513, the Gram-negative *N. oralis* 14F2 FG-15-7B, *F. nucleatum* ATCC 25586, and *V. parvula* DSM, the anaerobic *F. nucleatum* ATCC 25586 and *V. parvula* DSM 2008, and the total mixed bacteria from pooled saliva and supra- and subgingival plaques of volunteers were all treated and compared. aPDT-VIS-wIRA with LE as PS significantly (*p* < 0.008) reduced the growth of all tested Gram-positive, Gram-negative, as well as aerobic and anaerobic bacterial strains, whereas without irradiation no reductions were seen (*p* < 0.0001). NaCl, with or without irradiation, was ineffective. After treatment with CHX 0.2%, the highest killing rate (100%) was observed, and no bacteria (0 log10 CFU) were cultivable. The method also significantly reduced all of the bacteria present in saliva and in the gingival biofilms. Three-dimensional visualization of viable and non-viable microorganisms revealed that LE penetrated deeper into the cell wall layers than CHX 0.2%. LE was an appropriate PS for eradicating microorganisms with VIS-wIRA, either in their planktonic form or in saliva and gingival plaque biofilms. These results encourage further investigation in order to determine which LE compounds contribute to the photosensitizing effect and to evaluate the size of the effect on maintaining oral health.

## 1. Introduction

Natural single compounds or multicomponent mixtures have successfully been employed as photosensitizers (PSs) for antimicrobial photodynamic treatment (aPDT) with visible light plus water-filtered infrared-A irradiation (VIS-wIRA). The mechanism of aPDT involves the excitation of a photosensitizer to a high-energy triplet state. The activated photosensitizer then interacts with endogenous molecular oxygen, yielding reactive oxygen species (ROS) such as hydrogen peroxide, hydroxyl radical or superoxide ion (type I reaction). In a type II reaction, the activated photosensitizer interacts with molecular oxygen and leads to the production of highly reactive singlet oxygen. This mechanism of action leads to the targeted destruction of microbial cells [1]. aPDT-VIS-wIRA with toluidine blue (TB) or chlorine e6 eradicated initial and mature oral biofilms and also altered the surviving biofilm [2]. The oral pathogens *S. mutans* and *E. faecalis* were killed by a level of up to 2 log10. The elimination of planktonic salivary bacteria was greater than that of the bacteria located within the biofilms. All TB concentrations tested were highly effective [1]. The bactericidal effects of aPDT-VIS-wIRA were demonstrated on a number of pathogens, such as *Aggregatibacter actinomycetemcomitans*, *Porphyromonas gingivalis*, *Eikenella corrodens*, *Actinomyces odontolyticus*, *Fusobacterium nucleatum*, *Parvimonas micra*, *Slackia exigua*, and *Atopobium rimaea*, which have all been associated with periodontal diseases [3]. The aforementioned pathogens were also associated with periodontal diseases [4,5,6]. In addition, in situ subgingival biofilm bacteria were eradicated, which was confirmed by Live/Dead staining [3]. The quantification of the stained photoinactivated microorganisms confirmed these results. Overall, confocal laser scanning microscopy (CLSM) revealed the diffusion of the tested PSs into the deepest biofilm layers after exposure to aPDT-VIS+wIRA [1]. When St. John’s wort extract was used as the PS, all biofilms were completely eradicated. *H. perforatum* rinsed off prior to aPDT-VIS-wIRA killed more than 92% of the initial viable bacterial count and 13% of the mature oral biofilm, indicating that aPDT using VIS+wIRA with *H. perforatum* extract may be a promising alternative treatment of biofilm-associated oral disease [7]. Another option is the use of natural juices as the PSs. Up to 5 log10 of *S. mutans* and *S. sobrinus* were killed by aPDT-VIS-wIRA with 0.4% and 0.8% pomegranate juice, 3% and 50% chokeberry juice, and 12.5% bilberry juice (both strains). Concentrations of at least 25% (pomegranate) and >50% (chokeberry and bilberry) eradicated the mixed bacteria in the saliva samples. Natural pomegranate juice was superior to chokeberry and bilberry juices as a natural multicomponent PS for the in vitro killing of periodontal pathogens [8]. Although adjuvant systems in non-surgical peri-implant treatment were only associated with minor improvements in health status in support of mechanical debridement therapy, more studies would be needed to assess the benefit of these therapies, and in particular of aPDT-VIS+wIRA [9]. The latter has not yet been investigated clinically, but its benefit seems likely.

Lingonberry (*Vaccinium vitis-idaea*) is a low-bush plant growing in the northern hemisphere. The “superfruit” collected in the wild is richer in antioxidants (radical scavenging vitamins and polyphenols) than other berries, and provides not only potent antioxidant, anti-inflammatory and antimicrobial activities, but it has also been found to help prevent and treat cancer, aging of the brain, and neurodegenerative disorders [10]. Lingonberries contain flavonoids (e.g., quercetin derivatives, the monomeric flavanols catechin and epicatechin), oligomeric proanthocyanidins type A and B, phenolic acids (e.g., derivatives of ferulic acid, coumaric acid, caffeoylquinic acid, and benzoic acid), antho-cyanins (water-soluble pigments responsible for the red color, mainly cya-nidin-3-galactoside and its derivatives), organic acids, various triterpenoids, vitamin A, B1, B2, B3 and C, potassium, calcium, magnesium, and phosphorous [10]. The spectrum of compounds in lingonberry extract (LE) is affected by the region of harvest, the strain, the growing environment (e.g., weather, soil conditions), ripening stage and extraction methods [10]. The aim of this study was to evaluate if multicomponent LE was an appropriate PS for aPDT using VIS-wIRA.

## 2. Materials and Methods

### 2.1. Radiation Source and Photosensitizer

A broadband visible light (VIS) and water-filtered infrared-A (wIRA) radiator (Hy- 83 drosun 750 FS, Hydrosun Medizintechnik GmbH, Müllheim, Germany) containing a 7 mm water cuvette was used as previously described [2]. Furthermore, an accessory orange filter (BTE31) was adjusted to the radiator, more than doubling the weighted effective integral irradiance possible in terms of the absorption spectrum of protoporphyrin IX compared to a standard orange filter (BTE595) [1]. A wavelength range from 570 nm to 1400 nm, with local minima at 970 nm, 1200 nm and 1430 nm was covered by the continuous water-filtered spectrum, on behalf of the absorption of water molecules [11]. The absolute power density of the irradiation was 200 mW cm^−2^, while it was composed of 48 mW cm^−2^ VIS and 152 mW cm^−2^ wIRA. The distance from the samples to the light source was 20 cm and the irradiance was applied for 5 min. A 24-well plate carrying the samples was put into a 37 °C warm water bath, to prevent heating during the irradiation.

We used multi-component lingonberry extract (LE 77982, Alpinamed AG, CH-9306 Freidorf) as the PS. According to the manufacturer’s information, the water-soluble red powder contained 5.1% (*m*/*m*) procyanidins and 1.1 g vitamin C per 100 g. To prevent any light-induced chemical reaction, the LE was diluted in 0.9% saline to a final concentration of 0.5 g/mL, and stored in the dark at 4 °C for a maximum of 7 days. The absorbance spectrum (Figure 1) of LE was measured using a Tecan Infinite 200 reader (Tecan, Crailsheim, Germany). Using a dilution series, the concentrations 0.25 g/mL, 0.125 g/mL and 0.0625 g/mL were tested in the experiments.

### 2.2. Bacterial Strains

A representative selection of pathogenic bacteria of the oral cavity [12] was provided by the Institute of Medical Microbiology and Hygiene, Albert-Ludwigs-University, D Freiburg. These included the Gram-positive bacteria *E. faecalis* T9, *S. mutans* DSM20523, *S. oralis* ATCC 35037 and, *S. sobrinus* PSM 203513, the Gram-negative *N. oralis* 14F2 FG-15-7B, *F. nucleatum* ATCC 25586 and, *V. parvula* DSM 2008, as well as the anaerobic strains *F. nucleatum* ATCC 25586 and *V. parvula* DSM 2008. Long-term storage of the bacterial strains was performed at −80 °C in brain heart infusion medium containing 15% (*v*/*v*) glycerol for aerobic germs and in basal glucose phosphate growth medium containing 15% (*v*/*v*) glycerol for anaerobic germs [13]. Aerobic bacteria were cultivated on Columbia blood agar (CBA) plates at 37 °C in aerobic atmosphere with 5% CO_2_, as described earlier [3]. Yeast-cysteine blood agar (HCB) plates were used for cultivating anaerobic bacteria at 37 °C under anaerobic conditions with gaspack systems (BD GasPack™, Becton, Dickinson and Company, Franklin Lakes, NJ, USA) in airtight anaerobic containers (Anaerocult^®^ A, Merck, Darmstadt, Germany). The bacterial strains were subcultured weekly or defrosted from long-term storage. Preparing bacterial suspensions for the experiments, overnight cultures were grown in tryptic soy broth (Merck) in a shaking water bath (GLF 1086, Burgwede, Deutschland) at 37 °C. The following morning, the turbid solution was centrifuged at 3500× *g* for 10 min. The resulting pellets were resuspended in 5 mL NaCl for aerobic or in GC-medium for anaerobic bacteria. To adjust to a final bacterial concentration of 10^7^ germs/mL, the resuspended solution was diluted 1:100 in NaCl or GC-medium for the aerobic and anaerobic germs, respectively.

### 2.3. Total Human Salivary Bacteria, Supra- and Subgingival Plaque Samples

After approval of the study by the Ethics Committee of the Albert-Ludwigs-University of Freiburg, three healthy volunteers gave written consent to donate their saliva (Nr. 91/13). The exclusion criteria included: the use of antibacterial mouth rinses or antibiotics, participation in another clinical study three months prior to the start of the experiment, pregnancy, lactation, smoking, or severe systemic diseases. The volunteers donated unstimulated human saliva, which was pooled to gain total salivary bacteria. Pooled, unstimulated saliva was diluted in 0.9% NaCl for aerobic bacteria or GC-medium for anaerobic testing, with a dilution factor of 1:100. All total salivary samples were taken on the same day, directly before conducting the experiments, to avoid alteration of the total salivary microbiota.

The supragingival samples were collected from three caries patients, whereas the subgingival plaque samples were gained from three periodontitis patients. All patients gave their written consent to the study. The material was immediately frozen at −80 °C for storage. Prior to the experiment, the plaque samples were defrosted and pooled. Centrifugation at 6000× *g* for 10 min was performed for washing, to remove blood residues from the samples. The remaining solid parts were put in 2 mL RTF (reduced transfer fluid) medium [14]. Until the experiments, the pooled and portioned plaque samples were again stored at −80 °C. Right before starting the experiment, the samples were defrosted and diluted 1:100, in 0.9% NaCl for aerobic samples and in GC-medium for anaerobic germs.

### 2.4. APDT Protocol and Quantification

Next, 500 µL of LE diluted in 0.9% NaCl to the concentrations 0.25 g/mL, 0.125 g/mL and 0.0625 g/mL were prepared in duplicate on a multiwell plate (24-well plate, Greiner Bio-One; Frickenhausen, Germany). Subsequently, 500 µL of the different bacterial, saliva or plaque suspensions were added to each well. After an incubation time of 2 min in the dark, the 24-well plate was put into a preheated water bath (37 °C) under VIS-wIRA irradiation. Another multiwell plate was arranged in the same way but without radiation. For each plate, chlorhexidine CHX 0.2% served as a positive control and NaCl as a negative control. Every experiment was conducted twice.

A dilution series of radiated and non-radiated bacterial samples was carried out up to a dilution of 1:10^5^. Serial dilution ensured that the range of bacterial growth was covered and countable within the detection limit, which varied depending on the bacterial isolates. Next, 100 µL of each dilution was streaked onto agar-plates, followed by cultivation at 37 °C. For aerobic testing CBA-plates were incubated under an aerobic atmosphere of 5% CO_2_ for 2–3 days. Anaerobic germs were cultured on HCB-plates for 7 days under anaerobic conditions (see above). Saliva and plaque samples underwent aerobic, as well as anaerobic testing. After incubation, CFU (colony-forming units) were quantified with the aid of a colony counting device (WTW BZG 40, Xylem Inc., New York, NY, USA).

### 2.5. Live/Dead-Staining with Confocal Laser Scanning Microscopy (CLSM)

Live/Dead staining with subsequent confocal laser scanning microscopy was performed to determine the amount of living and dead bacteria after the application of aPDT with VIS+wIRA in combination with LE on plaque and saliva. Fluorescent SYTO^®^ 9 nucleic acid binding dye can penetrate intact cells and disrupt membranes. It was used in combination with fluorescent propidium iodide (PI) nucleic acid binding dye (Live/Dead Baclight™ Bacterial Viability Kit, Life Technologies GmbH, Darmstadt, Germany), which can only enter disrupted microorganisms [1]. In the dyed samples the viable bacteria fluoresce green, whereas only the non-viable bacteria fluoresce red.

LE diluted to a concentration of 0.25 g/mL was added to saliva, supra- and subgingival plaque samples on a 24-well-plate. The positive control was performed with CHX 0.2% without irradiation. NaCl served as a negative control, with and without irradiation. Each sample was analyzed before and after irradiation. The plate with irradiation was treated with VIS+wIRA after an incubation period of 2 min, as described above. All samples were collected in Eppendorf vessels and centrifuged at 6000× *g* for 10 min. The supernatant was removed and NaCl was added twice for washing out the remaining LE. The resulting pellet was dissolved again in NaCl before the dyes were added. SYTO^®^ 9 and PI solutions were combined at a ratio of 1:1, before 1 µL was added to the samples. Incubation followed for 10 min in the dark at room temperature. Centrifugation and piping off the supernatant left the dyed pellets. A fixation solution was prepared by cooling CygelTM (Biostatus Ltd., Shepshed, UK) on a block of ice for 2 min, after which 40xPBS was added to the cooled Cygel and mixed. The fixation solution was added to the pellets. While still being liquid, the fixation of the gel containing the prepared pellets, was pipetted onto 8-well slides (μ-Slide 8 well, ibidi GmbH, Munich, Germany). At room temperature, the gel quickly hardened and was then ready for analysis by confocal laser scanning microscopy (LSM 710, AxioObserver). Using a 40× objective (LD C-Apochromat 40×/1.1 Korr M27) each sample was scanned at three representative positions, creating a single, maximum projection image at each position, as well as a Z-stack made up of optical slices through the depth of the biofilm. The Z-stacks permitted 3-dimensional observation of the examined sections. For image acquisition, an average of 4 was used, with an overall resolution of 1024 × 1024 pixels, zoom factor of 1, and a pinhole setting of 1 airy unit for confocality.

### 2.6. Image Analysis

The images were analyzed using the Imaris software package (Nr. 9.9.1). Splitting the green and the red channels for each stack, the total volumes (surface) of viable and non-viable bacteria in the scanned areas were defined. Background noise was eliminated, excluding less intense signals by setting appropriate thresholds. The resulting sums of µm^3^ or µm^2^ were converted into percentages. Statistical calculation was then used to determine whether the applied aPDT-VIS-wIRA significantly decreased the percentage of viable bacteria. Furthermore, the analyzed images were photographed in the plane, as well as in a 3-dimensional view.

## 3. Results

### 3.1. aPDT-VIS-wIRA with LE Significantly Reduced Bacterial Counts

aPDT using VIS+wIRA in combination with LE as PS significantly (*p* < 0.008) reduced the growth of all tested Gram-positive, Gram-negative, as well as aerobic and anaerobic bacterial strains, whereas without irradiation there was no impact on bacterial growth.

Figure 2 presents the eradication rates of the tested bacterial strains. *Neisseria oralis* showed a dose-dependent reduction by a level of 6.6-5.1 log_10_ CFU, with no further increase at the highest LE concentration. A weaker effect was shown for *Streptococcus mutans* with a significant dose-dependent reduction of bacterial growth over 1.6 to 0.49 log_10_ CFU, and for *Streptococcus sobrinus* (2.4-1.2 log10 CFU). In contrast, aPDT using VIS-wIRA with LE on strains of *Enterococcus faecalis* (2.8-2.4 log_10_ CFU) and *Streptococcus oralis* (5.6-3.7 log_10_ CFU) caused a significant reduction in the bacterial count at each of the tested LE concentrations, which resulted in complete eradiation for *Streptococcus oralis* except for the lowest LE dose tested.

For anaerobic bacteria, such as *Veillonella parvula*, a significant dose-dependent reduction of bacterial growth was shown (2.6-0.47 log_10_ CFU) for the three dilutions, respectively. The anaerobic bacterium *Fusobacterium nucleatum* also underwent significant reductions of 4.7-3.5 log_10_.

### 3.2. aPDT-VIS-wIRA in Combination with LE Significantly Killed Microorganisms in Human Saliva and Oral Biofilm Samples

Figure 3 depicts the effects of aPDT in combination with LE on human saliva and oral supra- and subgingival biofilm samples. Adding LE without irradiation did not affect human salivary bacteria when compared to the negative control (NaCl). CHX 0.2% eradicated all salivary bacteria (0 log_10_ CFU). The number of aerobic salivary bacteria was significantly decreased by 3.4-1.9 log_10_ CFU after treatment with LE as PS in aPDT using VIS+wIRA. For the anaerobic salivary bacteria, a similar trend was shown with a significant reduction of 2.7 to 1.7 log_10_ CFU.

aPDT using VIS+wIRA in combination with LE on aerobic supragingival microorganisms were significantly reduced by >1.8 log_10_ CFU at all tested LE concentrations. Anaerobic supragingival plaque bacteria were also significantly eradicated by 2.4-1.6 log_10_ CFU. Likewise, applying aPDT-VIS+wIRA on subgingival plaque samples with LE as the PS significantly reduced aerobic microorganisms by 1.6-1.05 log_10_ CFU. There was a significant decrease in anaerobic subgingival bacteria over 2.2 to 1.4 log_10_ CFU.

### 3.3. Live/Dead Staining with CLSM, Visulization of Penetration Depth

Representative images of CLSM after live/dead staining are displayed in Figure 4. The viable bacteria were presented in green, whereas the dead bacteria were depicted in red. 3D-view, in addition to the cross-sectional presentation, allowed an evaluation of the penetration depth into the samples. Applying aPDT using VIS+wIRA in combination with LE to the samples significantly reduced the number of viable bacteria, corresponding to the red parts. 3D-view and observation of the CLSM z-section galleries presented efficient penetration of LE into deep layers of the analyzed samples. LE without irradiation did not affect bacterial growth, so the dyed bacteria remained primarily green. Similar images were shown for the negative controls using NaCl before and after irradiation. For the positive control with CHX (0.2%), a significant loss of viable cells was detected, but the effect remained more superficial compared to the effect of aPDT with VIS+wRIA in combination with LE.

## 4. Discussion

The antimicrobial effects of preparations from LE have been investigated in different studies, however, to date, these have not considered LE as a PS. In vitro studies reported that the active principle of lingonberry inhibited the growth of *Candida* species, *S. mutans*, *Porphyromonas gingivalis*, *Fusobacterium nucleatum*, *S. aureus*, *Salmonella enterica* sv *Typhimurium*, *S. epidermidis*, *P. gingivalis*, *P. intermedia*, the antiaggregation of *S. mutans* with *Fusobacterium nucleatum* or *Actinomyces naeslundii*, the anti-adhesiveness of *Neisseria meningitidis* or oral streptococci in biofilm formation, and the binding activity of *Streptococcus pneumoniae*, *Streptococcus agalactiae* and *Streptococcus suis* [10,15]. Polyphenols contribute to the lingonberry antibacterial activity; a polyphenol-rich fraction at 0.5–1 mg/mL significantly reduced biofilm formation by *Streptococcus mutans*, *S. sobrinus*, *S. sanguinis*, and reduced the bioactivity of *S. mutans* at 1–2 mg/mL [16]. In particular, tannins isolated from the lignonberry polyphenol fraction, such as procyanidin B-1, procyanidin B-3, proanthocyanidin A-1, cinnamtannin B1, epicatechin, and catechin, showed antimicrobial activity against selected periodontal pathogens, including *Porphyromonas gingivalis* and *Prevotella intermedia*. Epicatechin-(4beta → 8)-epicatechin-(4beta → 8, 2beta → O → 7)-catechin had the strongest antimicrobial activity against *P. gingivalis* and *P. intermedia*, with a MIC of 25 µg/mL, but was not effective against *A. actinomycetemcomitans*. The other tannins tested did not show antimicrobial activity [17]. While the combination of anthocyanins and co-pigments in the extract possessed the highest antioxidant activities, each sample (extract, anthocyanins as well as the co-pigment fraction) protected the cells from oxidative stress. Thus, synergistic effects between phenolic compounds seem to be responsible for the high antioxidant potential of lingonberries [18], which determines the antimicrobial activity.

Clinical studies showed that fermented lingonberry juice (FLJ) was beneficial for oral health. In 30 adults who used FLJ as a mouthwash twice daily for one week, *Streptococcus mutans* and *Candida* counts, visible plaque index, and bleeding upon probing were all reduced, while *Lactobacilli* counts had increased [19]. However, the expectations were not met in a pilot study on participants with dental implants: the difference between groups with and without FLJ mouthwash twice daily for 15 days diminished after the FLJ regimen had ended. Remarkedly, the decrease in aMMP-8 (active matrix metalloproteinase) levels appeared to continue [20]. This key biomarker for periodontitis and peri-implantitis had a sensitivity of 75–85% and a specificity in the range of 80–90% [21]. In a one-year prospective intervention study on the anticaries, anti-inflammatory, antiproteolytic, and antimicrobial effects of FLJ (6 months with, and 6 months without mouthwash) the levels of *S. mutans* and *Candida* counts, the decayed surfaces, bleeding on probing, and visible plaque indices had each decreased significantly during the FLJ period. *Lactobacilli* counts had increased significantly, while there was also a significant difference in aMMP-8 levels, decayed, missing, and filled teeth, and surfaces between the three measurement points. Probing pocket depths were, however, not affected [22]. An analysis of the subjective dry mouth sensation questionnaires revealed that FLJ had decreased symptoms of xerostomia. Once-a-day use of the FLJ mouthwash had increased salivary flow rates, buffering capacity, and salivary pH. Thus, there is no doubt that FLJ had beneficial effects on oral health [23].

The results of our study show that when diluted in non-antimicrobial concentrations, LE may serve as a PS for aPDT using VIS+wIRA. LE is, thus, another natural multicomponent PS option next to St. John’s wort extract [24] and the natural juices of pomegranate, chokeberry and bilberry [8]. Significant antibacterial activity was detected against a number of periodontal pathogenic bacterial strains, as well as against in situ saliva and gingival plaque biofilms (supra- and subgingival) with mixed bacteria. Planktonic bacterial load, irrespective of the isolate (Gram-positive or -negative, aerobic or anaerobic) was reduced by at least 97.9% (0.25 g/mL), which is considered to be sufficient for the eradication of oral pathogens [1,3]. Eradication rates decreased with lower concentrations of LE as shown in Figure 2 and Figure 3. At a concentration of 0.125 g/mL, more than 94.7% of the bacteria were still killed with regard to both single bacterial strains and pooled plaque, and saliva samples. When diluted to 0.0625 g/mL, bacterial growth was reduced by >91.1%, except for *Veillonella parvula* (reduction of only 66.4%) and *S. mutans* (reduction of only 67% at a concentration of 0.125 g/mL). The lowest concentration tested (0.0625 g/mL), did not significantly inhibit bacterial growth. A concentration of 0.25 g/mL LE seemed to be the optimum LE concentration for aPDT using VIS+wIRA. With a killing rate of 100%, CHX showed the highest antimicrobial activity.

Naturally occurring biofilms consist of a network of bacteria, which make them more resistant to antimicrobial treatments [25,26]. The new strategies to counteract the natural resistance/tolerance of microbial biofilms against antimicrobial agents and to mitigate their pathological effects can be classified into four main categories: (I) prevention, (II) weakening, (III) disruption, and (IV) killing [27]. Among the killing strategies, aPDT using VIS+wIRA with PSs has proven to be very effective. In our pooled saliva and sub- and supragingival plaque samples, a significant killing rate of at least 98% was demonstrated. LE without irradiation did not reduce bacterial growth. The bactericidal antibiofilm effect is, therefore, attributable to the photodynamic activation of the natural multicomponent PS LE. Not only the LE concentration, but also characteristics of the bacterium cell wall structure can impact the success of the treatment [28]. Small hydrophilic drugs, such as β-lactams, use pore-forming porins to obtain access to the cell interior, whereas macrolides and other hydrophobic drugs diffuse across this lipid bilayer [28]. It seems likely that the small compounds in LE may be converted from soluble to transmembrane forms, exposing sufficient hydrophobicity to drive spontaneous bilayer insertion [29]. As we have previously shown for natural multicomponent PSs, no single compound was responsible for the cell-damaging effects, which were more likely due to the administration of a mixture of compounds [10]. Our visualization after aPDT-VIS+wIRA of viable planktonic bacteria and bacteria in saliva and gingival plaques demonstrate that LE was able to penetrate into deeper bacterial wall layers than CHX 0.2%. It remains to be seen which compounds produce the photosensitizing activity, and whether oral aPDT using VIS+wIRA with LE is effective in the clinical setting of periodontitis and peri-implantitis. It should be emphasized that the broad-band light source of VIS-wIRA that was used has some advantages compared to single-wavelength laser units, wide-band halogen lamps and LED appliances [30,31]. These advantages comprise the flexibility of using VIS+wIRA for a wide range of PSs with different absorption maxima in addition to avoiding overheating of the tissue. Nevertheless, the fact that the juice only shows absorption in the 570 nm range may have weakened the efficiency of the applied aPDT. The photodynamic effects caused by the absorption of the lingonberry extract near 570 nm should be confirmed in future studies by measuring the radical oxygen species generated by illumination of lingonberry extract with VIS-wIRA. This is especially necessary because the emission spectrum of the light source that was used does not match the absorption maximum of lingonberry extract. Moreover, wound healing effects were reported for VIS-wIRA due to its mild thermal effect, which resulted in a higher perfusion level and increased tissue oxygen partial pressure [1,32]. Hence, additional clinical studies are needed to evaluate the impact of aPDT using VIS-wIRA in combination with LE on the healing effects of patients suffering from peri-implantitis or periodontitis.

The limitations of our study include (i) the discrepancy between the emission spectrum of the used light source and the absorption maximum of LE, (ii) the lack of identification of which compounds of the extract contributed to the overall antimicrobial effect of aPDT with VIS-wIRA with LE as the PS, and (iii) the small number of caries and peridontitis patients that contributed to the biofilm samples.

## 5. Conclusions

LE as a photosensitizing agent in aPDT revealed an encouraging antimicrobial activity in this study. Testing single bacterial strains, as well as in situ collected gingival plaque and saliva, we were able to show a significant reduction of bacterial growth. Still, the highest antimicrobial activity was shown by CHX 0.2%, with a killing rate of 100%. As LE is rich in polyphenols, it features antimicrobial, anti-inflammatory, and favorable tissue healing effects. Considering these health promoting factors along with the positive properties of the selected light source VIS+wIRA, the described procedure seems promising for further investigation. Precise analyses of LE’s composition, as well as clinical studies in vivo should be performed in the future.

## Figures and Tables

**Figure 1 nutrients-15-04988-f001:**
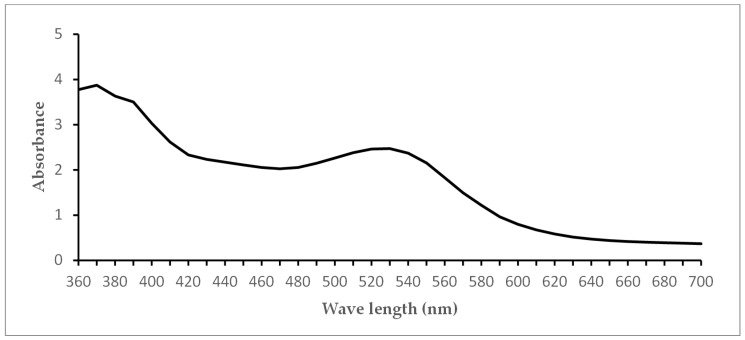
Absorbance spectrum of the lingonberry extract tested in this study.

**Figure 2 nutrients-15-04988-f002:**
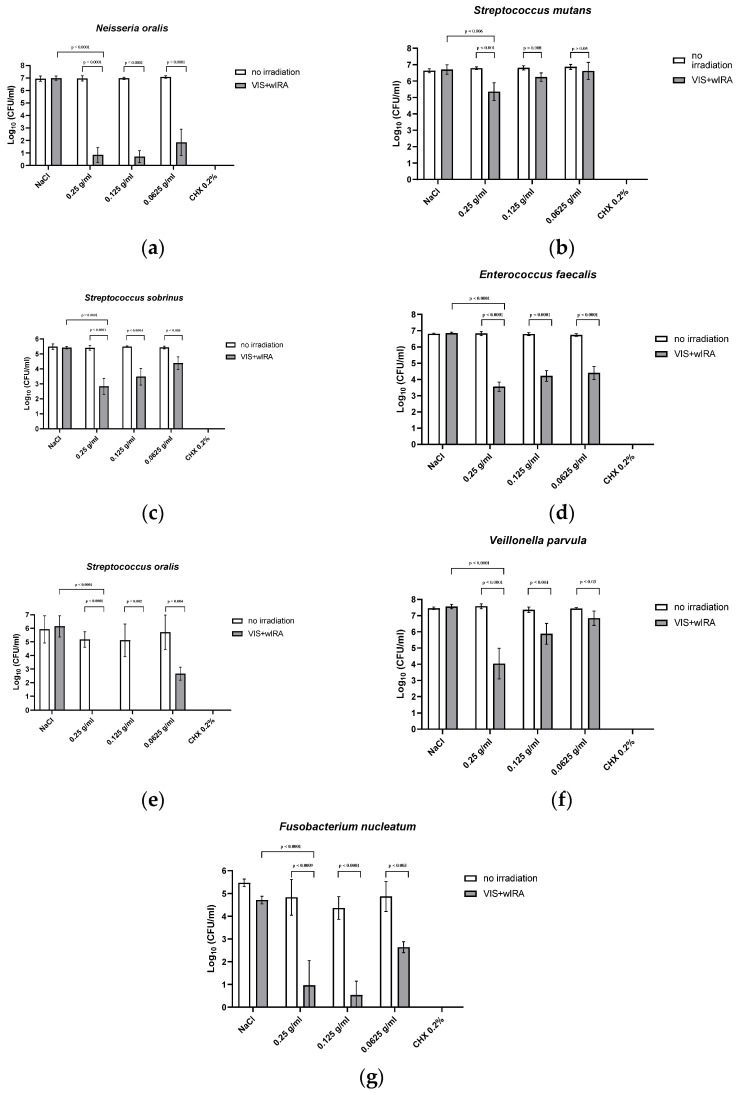
The effects of aPDT using VIS+wIRA in combination with LE on bacterial strains. (**a**) Neisseria oralis; (**b**) *Streptococcus mutans*; (**c**) *Streptococcus sobrinus*; (**d**) *Enterococcus faecalis*; (**e**) *Streptococcus oralis*; (**f**) *Veillonella parvula*; (**g**) *Fusobacterium nucleatum*.

**Figure 3 nutrients-15-04988-f003:**
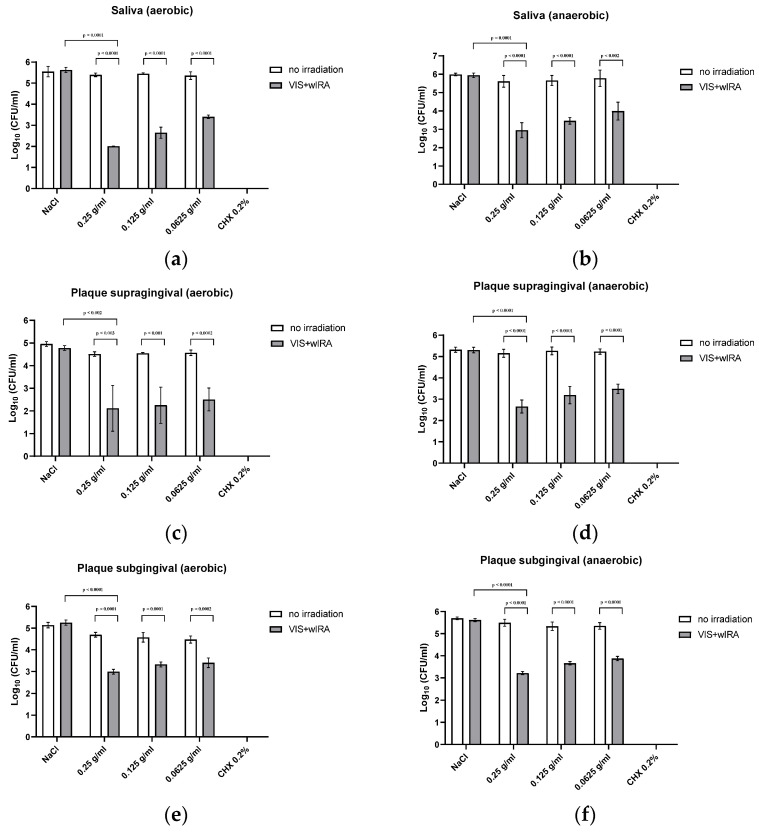
The effects of aPDT using VIS+wIRA in combination with LE on human saliva and oral supra- and subgingival biofilm samples. (**a**) Saliva aerobic testing; (**b**) Saliva anaerobic testing; (**c**) Supragingival plaque aerobic testing; (**d**) Supragingival plaque anerobic; (**e**) Subgingival plaque aerobic testing; (**f**) Subgingival plaque anaerobic testing.

**Figure 4 nutrients-15-04988-f004:**
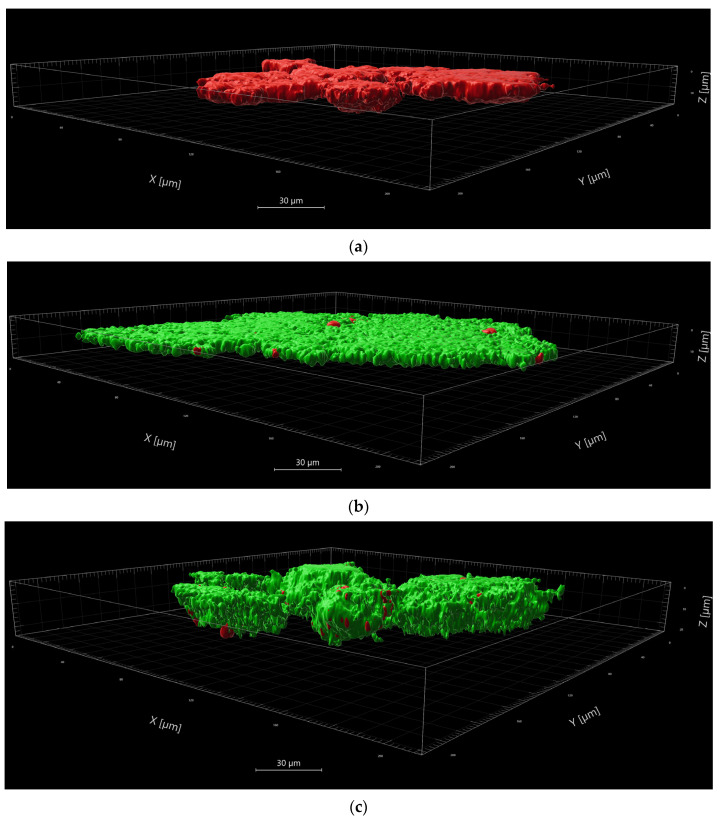
CLSM after Live/Dead Staining, 3D-reconstruction. Viable bacteria were depicted green, while dead bacteria appeared red. (**a**) Supragingival plaque after aPDT with VIS+wIRA and LE; (**b**) LE without irradiation; (**c**) NaCl as negative control; (**d**) CHX0.2% as positive control.

## Data Availability

Data available on request due to restrictions, e.g., privacy or ethical. The data presented in this study are available on request from the corresponding author. reported.

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
