# Peer review of "In Vitro Eradication of Planktonic, Saliva and Biofilm Bacteria Using Lingonberry Extract as a Photosensitizer for Visible Light Plus Water-Filtered Infrared-A Irradiation"

_nutrients, 2023, doi:10.3390/nu15234988_

Round 1

Reviewer 1 Report

Comments and Suggestions for Authors

Dear authors, the article is interesting but to be published it needs of some adjustment.

Title:

-  I suggest to improve the title making the reader know what type of study it is. For example: "in-vitro study".

Introduction:

-  You should improve your references adding also clinical article because all this knowledge about microbiology at the end have the role to guide the clinical activity and the clinical research.

-  Line 46-50:  Please in this sentence there are no citations, you should cite some other papers that mentioned about the antimicrobial proprierties of chemical agents in periodontal and peri-implant diseases. I suggest you to cite in this case this review and this clinical article.

Butera A, Maiorani C, Gallo S, Pascadopoli M, Venugopal A, Marya A, Scribante A. Evaluation of Adjuvant Systems in Non-Surgical Peri-Implant Treatment: A Literature Review. Healthcare (Basel). 2022 May 11;10(5):886. doi: 10.3390/healthcare10050886.

-  Line 71-77:  Also in this sentence there are no citations, you should cite some agents with similar proprierties such as probiotics.

-  The results are well reported, the discussion is very good.

-  Please in the discussion explain the limitation of your study too.

-  Please, make softer the conclusion

Author Response

Reviewer 1

Dear authors, the article is interesting but to be published it needs of some adjustment.

Response: Thank you for this comment.

  1. Title: I suggest to improve the title making the reader know what type of study it is. For example: "in-vitro study".

Response: The title has now been modified to: “In vitro-Eradication of Planktonic, Salivary and Biofilm Bacteria Using Lingonberry Extract as a Photosensitizer for Visible Light plus Water-Filtered Infrared-A Irradiation”

  1. You should improve your references adding also clinical article because all this knowledge about microbiology at the end have the role to guide the clinical activity and the clinical research.

Response: We added three citations [4-6] to the introduction in order to stress the clinical role of some of the bacteria which were associated with periodontitis: “pathogens, such as Aggregatibacter actinomycetemcomitans, Porphyromonas gingivalis, Eikenella corrodens, Actinomyces odontolyticus, Fusobacterium nucleatum, Parvimonas micra, Slackia exigua, Atopobium rimaea [3]. The aforementioned pathogens were also associated with periodontal diseases [4-6]

  1. Line 46-50: Please in this sentence there are no citations, you should cite some other papers that mentioned about the antimicrobial proprierties of chemical agents in periodontal and peri-implant diseases. I suggest you to cite in this case this review and this clinical article.

Butera A, Maiorani C, Gallo S, Pascadopoli M, Venugopal A, Marya A, Scribante A. Evaluation of Adjuvant Systems in Non-Surgical Peri-Implant Treatment: A Literature Review. Healthcare (Basel). 2022 May 11;10(5):886. doi: 10.3390/healthcare10050886.

Response: The following text has now been added by citing the suggested review Article:  “Although adjuvant systems in non-surgical peri-implant treatment were only associated with minor improvements in the state of health in support of mechanical debridement therapy, more studies would be needed to assess the benefit of these therapies, and in particular of aPDT-VIS+wIRA [9]. The latter has not yet been investigated clinically, but its benefit seems likely”.

  1. Line 71-77: Also in this sentence there are no citations, you should cite some agents with similar proprierties such as probiotics.

Response: A reference has now been added at the end of the mentioned sentence: “Lingonberries contain flavonoids (e.g. quercetin derivatives, the monomeric flavanols cat-echin and epicatechin), oligomeric proanthocyanidins type A and B, phenolic acids (e.g. derivatives of ferulic acid, coumaric acid, caffeoylquinic acid, and benzoic acid), antho-cyanins (water-soluble pigments responsible for the red color, mainly cya-nidin-3-galactoside and its derivatives), organic acids, various triterpenoids, vitamin A, B1, B2, B3 and C, potassium, calcium, magnesium, and phosphorous [10].  

  1. The results are well reported, the discussion is very good

Response: Thank you for this positive comment.

  1. Please in the discussion explain the limitation of your study too

Response: The following text has now been added to the discussion:

Limitations of our study include (i) the discrepancy between the emission spectrum of the used light source and the absorption maximum of LE, (ii) the lack of identification of which compounds of the extract contributed to the overall antimicrobial effect of aPDT with VIS-wIRA with LE as PS, and (iii) the small number of caries and peridontitis patients that contributed to the biofilm samples.

  1. Please, make softer the conclusion

Response: We replaced “LE has proven to be a potent PS” by “LE was an appropriate PS”

“LE was an appropriate PS for eradicating microorganisms with VIS-wIRA, either in their planktonic form, or in saliva and gingival plaque biofilms. These results encourage the further investigation of which LE compounds contribute to the photosensitizing effect, and to evaluate the effect size in maintaining oral health.”

Reviewer 2 Report

Comments and Suggestions for Authors

Dear authors,

Although the idea of the article is interesting, there are some fundamental issues concerning the performance of the experiment:

1.     It is stated that the wavelength range of the VIS-wIRA device is from 570nm to 1400nm. However, in Figure 1, only the 360-700nm range is exhibited. Why is that? The tested absorbance of the PS is incomplete. Also in Figure 1, it is clear that the absorbance is higher at the lower wavelengths where the device doesn’t illuminate.

2.     Based on the mechanism of aPDT and the Jablonski diagram, the wavelength of the applied light should match with the absorption maxima of the respective photosensitizer. Here, according to Figure 1 and the VIS-wIRA device used, the “correct” wavelength range is from 570 to 600nm, since after that the absorbance is diminished. As such, the fluence of 14.4 J/cm2 applied (calculated from the written irradiance of 48mW/cm2 for 5mins) is spread and the exact amount of energy delivered for the 570-600nm is not known. Hence, the phenomenon of aPDT here is debatable.

3.     Planktonic bacteria strains were examined. These are not of clinical importance, since the microbiota are organized in biofilms in the oral cavity. Only the saliva samples (biofilms) are worth to be assessed, and these were donated by only three humans. This is a very low sample size and no robust results can be held.

4.     According to the results, the use of chlorhexidine 0.20% presents the best outcome. This is not so clear throughout the Results and Discussion parts, and it should be definitely stated at the Conclusion (and also the conclusion of the abstract), not to be misleading.

As for the manuscript is concerned, there are also some issues:

·      At the introduction, please state within 2-3 sentences the mechanism of aPDT.

·      At 2.1. subchapter it is written: “A broadband visible light (VIS) and water-filtered infrared-A (wIRA) radiator (Hy- 83 drosun 750 FS, Hydrosun Medizintechnik GmbH, Müllheim, Germany) containing a 7 84 mm water cuvette was used as described elsewhere in detail [1–6,8].”

Briefly explain what is described elsewhere. The reader may not have access to these references.

·      mW/cm2 is the unit for power density or irradiance, not power. Please correct throughout the manuscript.

·      At 2.3 subchapter it is written: “three healthy volunteers gave written consent to donate their saliva” and afterwards “three supragingival and three subgingival plaque samples were collected from six patients”.

Clarify the number of patients at the second sentence.

·      At Discussion it is written: “These advantages comprise the flexibility of using VIS+wIRA for a wide range of PSs with different absorption maxima in addition to avoiding overheating of the tissue.

Compared to lasers, this statement is not correct. Monochromaticity is a unique and inherent characteristic that provides the laser with the possibility to interact with the photosensitizer by accurately matching its peak absorption. This results in less excess energy and tissue heating, which is sub-optimal in delivering the PDT reaction, when compared to the effects of broad bandwidth devices.

Comments on the Quality of English Language

Minor editing is needed.

Author Response

Reviewer 2

  1. It is stated that the wavelength range of the VIS-wIRA device is from 570nm to 1400nm. However, in Figure 1, only the 360-700nm range is exhibited. Why is that? The tested absorbance of the PS is incomplete. Also in Figure 1, it is clear that the absorbance is higher at the lower wavelengths where the device doesn’t illuminate.

Response: Since the juice showed almost no absorption above 640 nm, it was not necessary to measure the absorption spectrum of lingonberry extract at higher than 700 nm. We agree with the reviewer that the peak absorption of the juice does not exactly match the excitation spectrum of the VIS-wIRA. Nevertheless, the absorption range of lingonberry extract at 570 nm was still sufficient to show a photodynamic effect. We have added the following text for discussion: “The fact that the juice only shows absorption in the 570 nm range may have weakened the efficiency of the applied aPDT.”

  1. Based on the mechanism of aPDT and the Jablonski diagram, the wavelength of the applied light should match with the absorption maxima of the respective photosensitizer. Here, according to Figure 1 and the VIS-wIRA device used, the “correct” wavelength range is from 570 to 600nm, since after that the absorbance is diminished. As such, the fluence of 14.4 J/cm2 applied (calculated from the written irradiance of 48mW/cm2 for 5mins) is spread and the exact amount of energy delivered for the 570-600nm is not known. Hence, the phenomenon of aPDT here is debatable.

Response: We agree with the reviewer that the optimum of absorption from the juice does not quite match with the spectrum of VIS-wIRA and that the photodynamic effect should be confirmed by studying the mechanism of the aPDT using lingonberry extract and VIS-wIRA. The following text has now been added to the discussion: “The photodynamic effects caused by the absorption of the lingonberry extract near 570 nm should be confirmed in future studies by measuring the radical oxygen species generated by illumination of lingonberry extract with VIS-wIRA. This is especially necessary because the emission spectrum of the light source which was used does not match the absorption maximum of lingonberry extract.”

  1. Planktonic bacteria strains were examined. These are not of clinical importance, since the microbiota are organized in biofilms in the oral cavity. Only the saliva samples (biofilms) are worth to be assessed, and these were donated by only three humans. This is a very low sample size and no robust results can be held.

Response: We agree with Reviewer 2 that clinical samples should be tested first and foremost. Nevertheless, since different bacterial species may have different response to the application of aPDT, it is useful to screen some planktonic bacteria to have an idea about the efficiency of the used photosensitizer in combination with the light source. As we also tested clinical samples of three volunteers and patients (total salivary bacteria, supra- and subgingival plaque), we believe that we have delivered a proof of principle about aPDT using VIS-wIRA and lingonberry extract. (Please see also Refernce Nr. 8: Chrubasik-Hausmann S et al., Nutrients. 2021 Feb 24;13(3):710. doi: 10.3390/nu13030710.)

  1. According to the results, the use of chlorhexidine 0.20% presents the best outcome. This is not so clear throughout the Results and Discussion parts, and it should be definitely stated at the Conclusion (and also the conclusion of the abstract), not to be misleading.

 Response: This point has now been emphasized throughout the manuscript by adding the following sentence: “With a killing rate of 100%, CHX showed the highest antimicrobial activity.”

  1. At the introduction, please state within 2-3 sentences the mechanism of aPDT.

Response: The following text has now been added to the introduction: “The mechanism of aPDT comprises the excitation of a photosensitizer to a high-energy triplet state. The activitaed photosensitizer interacts with the endogenous molecular oxygen yielding reactive oxygen species (ROS) such as hydrogen peroxide, hydroxyl radical or superoxide ion (type I reaction). In a type II reaction, the activated photosensitizer interacts with molecular oxygen and leads to the production of highly reactive singlet oxygen. This mechanism of action leads to the target destruction of microbial cells [1].

  1. At 2.1. subchapter it is written: “A broadband visible light (VIS) and water-filtered infrared-A (wIRA) radiator (Hy- 83 drosun 750 FS, Hydrosun Medizintechnik GmbH, Müllheim, Germany) containing a 7 84 mm water cuvette was used as described elsewhere in detail [1–6,8].”

Response: Actually, the description of the used light source was described in the present study in the subsequent text in this subchapter. The above mentioned text was rephrased as follows: “A broadband visible light (VIS) and water-filtered infrared-A (wIRA) radiator (Hy- 83 drosun 750 FS, Hydrosun Medizintechnik GmbH, Müllheim, Germany) containing a 7-mm water cuvette was used according to earlier own studies [2].”

  1. mW/cm2 is the unit for power density or irradiance, not power. Please correct throughout the manuscript.

Response: This term has now been corrected throughout the manuscript.

  1. At 2.3 subchapter it is written: “three healthy volunteers gave written consent to donate their saliva” and afterwards “three supragingival and three subgingival plaque samples were collected from six patients”.

Clarify the number of patients at the second sentence.

Response: The text has now been corrected as follows: “The supragingival samples were collected from three caries patients, whereas the subgingival plaque samples were gained from three periodontitis patients. All patients gave their written consent to the study.”

  1. At Discussion it is written: “These advantages comprise the flexibility of using VIS+wIRA for a wide range of PSs with different absorption maxima in addition to avoiding overheating of the tissue.”

Compared to lasers, this statement is not correct. Monochromaticity is a unique and inherent characteristic that provides the laser with the possibility to interact with the photosensitizer by accurately matching its peak absorption. This results in less excess energy and tissue heating, which is sub-optimal in delivering the PDT reaction, when compared to the effects of broad bandwidth devices.

Response: The reviewer is correct, when a monochromatic laser light source would be compared with the total visible light delivered by a conventional broad bandwidth device.  The spectrum of VIS+wIRA does not contain all the components of visible light, since the water lens eliminated some components. These components include Infrared B, Infrared C, parts of Infrared A and of course, the near-UV wavelengths. Hence, overheating was indeed avoided (Please see references 1, 11 and 32).

Round 2

Reviewer 2 Report

Comments and Suggestions for Authors

Dear authors,

The critical flaws regarding the methodology still remain, but they are explained at the text. Thus, the manuscript is accepted for publication.

Comments on the Quality of English Language

Minor editing is required.